# Permanent Spreading of 1RS.1AL and 1RS.1BL Translocations in Modern Wheat Breeding

**DOI:** 10.3390/plants12061205

**Published:** 2023-03-07

**Authors:** Varvara A. Korobkova, Ludmila A. Bespalova, Aleksey S. Yanovsky, Anastasiya G. Chernook, Pavel Yu. Kroupin, Andrey V. Arkhipov, Anna I. Yurkina, Lubov A. Nazarova, Aleksandra A. Mudrova, Anastasiya D. Voropaeva, Olga Yu. Puzyrnaya, Elena V. Agaeva, Gennady I. Karlov, Mikhail G. Divashuk

**Affiliations:** 1All-Russia Research Institute of Agricultural Biotechnology, 127550 Moscow, Russia; 2P.P. Lukyanenko National Grain Centre, Department of Breeding and Seed Production of Wheat and Triticale, Central Estate of KNIISH, 350012 Krasnodar, Russia

**Keywords:** wheat, rye, triticale, chromosomal rearrangements, PCR markers, KASP markers, in situ hybridization

## Abstract

Wheat-rye translocations 1RS.1BL and 1RS.1AL are used in bread wheat breeding worldwide because a short arm of rye chromosome 1 (1RS) when introgressed into the wheat genome confers resistance to diseases, pests and better performance under drought-stress conditions. However, in durum wheat genotypes, these translocations occur only in experimental lines, although their advantages could enhance the potential of this crop. P.P. Lukyanenko National Grain Centre (NGC) has successfully developed commercially competitive cultivars of bread and durum wheat demanded by many agricultural producers in the South of Russia for decades. Here, 94 accessions of bread and 343 accessions of durum wheat, representing lines and cultivars from collection, competitive variety trials and breeding nursery developed at NGC were screened for 1RS using PCR markers and genomic in situ hybridization. The 1RS.1BL and 1RS.1AL translocations were detected in 38 and 6 bread wheat accessions, respectively. None of the durum wheat accessions showed translocation, despite the fact that some of them had 1RS.1BL donors in their pedigree. The absence of translocations in the studied durum wheat germplasm can be caused by the negative selection of 1RS carriers at different stages of the breeding process due to low quality and difficulties in transferring rye chromatin through wheat gametes.

## 1. Introduction

The exploitation of the genetic potential of rye (*Secale cereale* L.) in the breeding of bread wheat is one of the reliable and effective ways of its improvement. The 1RS.1BL wheat-rye translocation has been widely used in the development of commercial varieties of bread wheat worldwide [1,2,3,4,5,6]. On the website developed by R. Schlegel, 1034 wheat accessions with 1RS.1BL translocation are listed [2]. Translocation 1RS.1BL involving 1RS from Petkus rye was transferred, in particular, in the bread winter wheat cultivars Avrora, Kavkaz and Skorospelka 35 from German line Neuzucht by a Soviet breeder P.P. Lukyanenko [3,7]. Kavkaz and Avrora have been widely used in breeding programs in Russia, Ukraine, Hungary, former republics of Yugoslavia, Bulgaria, the Czech Republic, Slovakia and other countries [3,8,9,10,11,12,13,14]. The 1RS.1BL translocation was also independently produced at the Lukyanenko National Grain Centre (NGC) and was introduced into wheat cultivars Polovchanka, Knyazhna and Krasota [15]. Translocation 1RS.1AL involving 1RS from Argentinian rye Insave was developed in cv. Amigo mostly from which it was transferred to other bread wheat cultivars [3,7]. 1RS.1AL occurs much rarer than 1RS.1BL and is mentioned in 101 accessions only in the list by R. Schlegel [2].

The popularity and wide distribution of 1RS in wheat cultivars are due to the cluster of genes that confer adaptivity and tolerance to biotic and abiotic stresses. Genes for resistance to leaf rust (*Lr26*), stem rust (*Sr31*), powdery mildew (*Pm8*), yellow rust (*Yr9*), wheat streak mosaic virus (*Wsm*), aphids (*Gb2* and *Gb6*), wheat curl mite (*Cmc3*) and others are localized to the short arm of 1R [16]. However, according to some reports, to date the resistance determined by these genes has lost its effectiveness as new virulent pathogenic biotypes that can infect wheat plants with the *Lr26*, *Pm8*, *Sr31* and *Yr9* have appeared in Europe, Asia and Africa [17,18,19,20,21,22,23,24,25]. Besides the partially overcome resistance to biotic factors, the presence of a short arm of chromosome 1R in the wheat genome results in better water supply, increased zinc efficiency, stronger root development and higher yields under water-stress environments compared to parental lines [26,27,28,29,30,31].

The presence of the 1RS.1BL translocation in the genome of bread wheat has a negative effect on baking quality [32,33,34]. This could be explained by the presence of rye genes coding for ω- and 40 K γ-secalins of the gliadin group encoded by the *Sec-1* and, presumably, *Sec-N* genes, respectively, and changing the fractional composition of the protein, reducing the volumetric yield of bread, the sedimentation index and rheological test properties [35,36,37,38]. Studies of the effect of the 1RS.1AL translocation on quality scores have shown that its presence does not lead to such a dramatic decrease in these scores compared to 1RS.1BL [39,40]. The low technological quality of flour and dough is also associated with the loss of low-molecular-weight glutenins encoded by locus *Glu-3*, located on the short arm of chromosome 1B of wheat [41,42]. However, in bread wheat the detrimental effect of rye translocation on the bread-making quality and dough properties can be compensated by desirable high-molecular-weight glutenins at *Glu-D1* locus [43,44].

In general, information on the active distribution and use of translocations from the rye genome in durum wheat breeding is practically absent in publicly available studies and datasets. However, tetraploid wheats with the 1RS.1BL translocation can be obtained by crossing bread wheat carrying1RS with durum wheat [45]. In Indian durum wheat varieties, the 1RS.1BL translocation did not change the protein content in the grain, but significantly reduced the SDS volume [46]. A comparison of eight isogenic durum wheat lines grown under Mediterranean conditions showed that 1RS.1BL increased protein content but decreased SDS volume [28].

Therefore, the question arises: what is the current status of 1RS.1BL and 1RS.1AL translocations and their place in the contemporary breeding process in bread and durum wheat? The analysis of the presence of 1RS.1AL and 1RS.1BL translocations in cultivars of bread and durum wheat, as well as promising lines from the competitive variety trial nursery (CVT, prefinal stage of cultivar assessment preceding its state variety trial, SVT, followed by the registration), developed at NGC, may provide an answer on this question.

## 2. Results

### 2.1. Detection of 1RS Using PCR Markers

Totally, 94 accessions of winter bread wheat and 343 accessions of winter and spring durum wheat were analyzed for the presence of 1RS.1BL and 1RS.1AL translocations using SSR marker SCM9 and KASP marker 1B1R_6110 (Appendix A). As a result of genotyping 63 winter bread wheat cultivars, translocation 1RS.1BL was shown in the following 33 cultivars: Adel’, Afina, Alekseich, Antonina, Avrora, Ayvina, Bagrat, Bezostaya 100, Gurt, Irishka, Karavan, Kavkaz, Knyazhna, Kollega, Kurs, Ol’khon, Polovchanka, Stan, Step’, Tanya, Urup, Utrish, Vanya, Vassa, Vekha, Velena, Vid, Videya, Vita, Viza, Vostorg, Yuka and Zhiva (Appendix A). None of the studied winter bread wheat cultivars carried translocation 1RS.1AL. Out of 26 promising lines of winter bread wheat from the CVT nursery, 1RS.1BL and 1RS.1AL translocations were found in five and six accessions, respectively (Figure 1 and Figure 2, Appendix A).

Application of both SCM9 and 1B1R_6110 markers did not reveal either 1RS.1BL or 1RS.1AL in any of the 94 durum cultivars (Appendix A) and 105 durum wheat lines from the CVT nursery, though some of them have bread wheat cultivars with the translocation in their pedigree (Appendix A). For example, lines 4306h15, 4320h38, 3902h3-18-31, 4517h25 were developed by one- or two-step crossing 1RS.1BL-carriers Zhiva, Alekseich, Vostorg and Utrish, respectively, with durum cultivars.

The absence of 1RS was also observed among all 144 lines of the durum wheat from breeding nursery, although they were at the very beginning of the breeding process. Moreover, all 144 lines analyzed were obtained from crosses (*T. aestivum* (+1RS.1BL) × *T. durum*) × *T. durum* with Alekseich, Yuka, Ayvina and Bezostaya 100 used in their pedigrees (Appendix A).

### 2.2. Cytogenetic Identification of 1RS Translocation

Genomic in situ hybridization (GISH) revealed centric translocation of the short arm of rye chromosome attaching to the long arm of wheat chromosome in cultivars Avrora, Vassa, Knyazhna, Polovchanka and Tanya as well as in the breeding line 2318h9-4-28 (Figure 3 and Figure 4).

## 3. Discussion

According to our and published data, translocation 1RS is common among common wheat cultivars developed at P.P. Lukyanenko NGC [13,20,47,48]. Using molecular markers, 44 out of 94 studied bread wheat accessions were shown to carry the 1RS chromatin: translocation 1RS.1BL was found in 33 commercial cultivars and five promising breeding lines, while 1RS.1AL was revealed in six breeding lines. NGC is the leading organization in wheat adaptive breeding, the largest breeding and technological centre of the North Caucasus and the Krasnodar Krai [15]. In just the last five years, for example, twenty-seven cultivars of winter bread wheat, two cultivars of winter durum and four cultivars of spring durum wheat developed at NGC were included in Russian State Register of Breeding Achievements. More than 8 and 6 million hectares of cultivated land in Russia and abroad, respectively, are occupied with wheat cultivars developed at NGC. Krasnodar Krai, with its fertile black soils (chernozems) in the flat part, is the leader in the gross grain harvest of wheat in Russia, while the high climatic potential allows increasing the harvest by an additional 1–3 million tons [49]. The climatic conditions of the Krasnodar Krai are characterized by great diversity due to the fact that it is located on the border of two climatic zones, temperate and subtropical; most of the territory of the region belongs to the climate of the steppe zone. Its characteristic features are continentality: hot, dry summers, little snowy winters with frequent thaws. The climate of the central zone of the Krasnodar Territory, where NGC is located, is characterized by cyclical and unstable weather: strong winds in the absence of snow cover, sharp temperature changes in winter, soil and atmospheric drought [50]. Changeable weather conditions provide a favorable background for the breeding of highly adaptive genotypes of winter wheat, adapted to winter freezing, drought and other unfavorable factors.

In cultivars Avrora, Vassa, Knyazhna, Polovchanka and Tanya with 1RS.1BL and in line 2318h9-4-28 with 1RS.1AL, the translocations were confirmed using GISH, the translocated chromosomes resulting from a fission–fusion event in the centromeric region. The GISH technique, in contrast to the PCR markers, allows for discriminating between plants homo- and heterozygous for the 1RS translocation. However, GISH does not show differences between 1RS.1BL and 1RS.1AL translocations as their size and arm ratio are quite similar (Figure 4). For cytogenetic differentiation between them, a FISH procedure with probes such as pSc119.2 and pAs1 should be applied [51,52,53].

Revealed 1RS.1BL and 1RS.1AL rearrangements are typical centric breakage-fusion chromosomes demonstrating stable mitotic stable chromosomes capable of normal behavior in cell divisions in common wheat [54]. Apparently, the stability of 1RS.1BL is due to the fact that the wheat centromere is inactivated and only the rye-derived centromere part incorporates CENH3, since centromeres with several active centromere units may have increased fragility and may result in new breakage and telocentric formation [55,56]. Thus, the centric type of 1RS chromosomal rearrangement remains the most common among commercial varieties. Despite the presence of a whole arm of the rye chromosome, which affects the quality of bread, breeding does not lead to a decrease in the size of rye chromatin in the translocated chromosome, and cultivars of bread wheat with 1RS translocation are in demand among producers.

Avrora and Kavkaz are cultivars involved in the pedigrees of many wheat varieties as a 1RS.1BL donor [3,8,9,10,11,12,13,14]. This translocation is widely used in the breeding programs at P.P. Lukyanenko National Grain Centre [57]. Moreover, an independent new translocation 1RS.1BL through the “triticale bridge” was developed at NGC in cultivars Polovchanka, Knyazhna and Krasota using one genetic stock, as well as in Tanya using another one [15,58]. Polovchanka was used in the pedigrees of the Veda and Yuka, Knyazhna in the pedigrees of Kuren’ and Vershina, Tanya in the pedigrees of Ol’khon and Gurt, all of the derived cultivars carrying 1RS.1BL. Thus, 1RS of alternative origin is widely involved in the breeding process of winter bread wheat, which contributes to the expansion of the genetic diversity of this crop [52,53].

Thus, it can be concluded that translocation 1RS.1BL is one of the most important components of the genotypic landscape for breeding high-yielding and highly adaptive winter bread wheat. Despite reports of overcoming resistance conferred by the genes located on 1RS [17,18,19,20,21,22,23,24,25], in general, the adaptability provided by this translocation into the wheat genome is still a highly demanded factor in the breeding of bread wheat. The presence of the 1RS.1BL was shown in cultivar Viza, which, according to the results of multi-year trials, showed high resistance to yellow stem rust and was classified as highly resistant, along with other carriers of the 1RS.1BL translocation Alekseich, Ayvina and Urup [47].

It is worth noting that a large number of lines from CVT nurseries were found to carry the 1RS.1AL translocation, which had not been detected in the registered commercial cultivars before our study. This is due to the expansion of the use of prebreeding material and the introduction of new sources of genes into the breeding process at NGC. Cultivars of other Russian breeding centers at the southern regions of Russia such as Bogdanka and Knyaginya Ol’ga [20,59] as well as Ukrainian cultivars such as Ekspromt, Kolumbiia, Smuglyanka and others [36,60], also carry the 1RS.1AL translocation, which indicates the prospects for using it in the breeding of winter bread wheat, adapted for growing in the conditions of the southern Russia.

None of 343 analyzed accessions of durum wheat, developed at NGC, demonstrated the presence of either 1RS.1BL or 1RS.1AL translocation despite the fact that they represent different stages of the breeding process and that some of them were developed using bread wheat carriers of translocations. The absence of translocations was observed both among 94 cultivars of the collection nursery and among 105 lines of the CVT nursery. None of the CVT lines, which have 1RS-carriers such as Zhiva, Alekseich, Vostorg and Utrish in their pedigree, inherited the translocation from them. It is also of interest that no translocations were detected in any of the 144 lines of the breeding nursery, although all lines were obtained as a result of crosses of the type (*T. aestivum* (+1RS.1BL) × *T. durum*) × *T. durum*, namely using 1RS.1BL donors such as Alekseich, Yuka, Ayvina and Bezostaya 100. In the studies of durum wheat cultivars of Hungary, Turkey and the Russian-Kazakhstan KASIB program, conducted by other authors, no translocations 1RS.1BL and 1RS.1AL were revealed as well [61,62,63].

The reason for the absence of 1RS.1BL and 1RS.1AL translocations among durum wheat accessions may be the selection against translocated lines at the early stages of the breeding process. At the breeding nursery, where durum wheat lines are positively selected for the amber kernel color and high yellowness index (YI), the lines derived from crosses with bread wheat usually show red kernel color and low YI, and, hence, are rejected. At the control nursery, these lines are rejected as they show gray to cream color at hearth bread baking test. At the final stage of the breeding process, at CVT, such breeding lines had low pasta strength, from gray to cream pasta color, high weight increase index. Altogether, these parameters result in low overall pasta quality score and, consequently, to negative selection. Probably, durum wheat has no genetical and biochemical mechanisms compensating the detrimental effect of 1RS on quality unlike bread wheat where the compensating function can be performed by the D subgenome, which plays a leading role in ensuring the quality of gluten [43,44].

In common wheat cultivars, the 1RS.1BL and 1RS.1AL translocation demonstrates transmission through gametes and stable heritability. The lack of fixation of 1RS.1BL and 1RS.1AL in the breeding process of durum wheat can be associated both with the problems in the formation of gametes with translocation during meiosis, with a decreased male fertility and greater competitiveness of gametes with the native complete chromosomes 1B and 1A, which lead to the elimination of the translocations [64]. The genome of tetraploid durum wheat is known to be generally more difficult to enclose interspecific chromosomal rearrangements than hexaploid common wheat [65].

Despite the absence of 1RS translocation in commercial cultivars of durum wheat, its potential and possible role in breeding for resistance to phytopathogens and tolerance to water-stressed conditions should not be underestimated. Studies have shown that genes for root development and water-stress resistance are localized in the distal region of 1RS [29,30]. The durum wheat lines with the 1RS translocation showed higher yields than the parental lines under dry conditions [28]. Rust and powdery mildew resistance genes are clustered between the *Sec-1* and *Sec-N* secalin genes [30,36]. Thus, in the future, marker-assisted selection and chromosomal engineering may help to reduce the dose of rye 1RS translocation in durum wheat by “preserving” rye chromatin fragments carrying genes for resistance to diseases and water stress, and “returning” the wheat gene for low-molecular-weight glutenins *Glu-3*, having a decisive influence on the quality of gluten [66,67]. Alternatively, it is also possible to use 1RS translocations with deletions for secalin genes [42].

The application of two marker systems showed identical results, and it also revealed the advantages and disadvantages of each of the systems: SCM9 is a dominant marker that can detect both 1RS.1AL and 1RS.1BL translocations, and the KASP marker 1B1R_6110 is a codominant marker, but it does not show the presence of the 1RS.1AL translocation, which can be critical for the marker-assisted breeding.

Screening germplasm collections and breeding lines for the presence of wheat-rye translocations using PCR-based markers at the initial stages of the breeding process can serve as one of the stages of wheat MAS-breeding to improve adaptive properties and enable breeders to perform precise selection of genotypes with or without this translocation at the development of the variety. An appropriate MAS-based choice of the parental pairs for crossing will make it possible to compensate for the negative effect of translocations in the future cultivars. In addition, tracking the desired traits early in the breeding process using molecular markers will significantly reduce the amount of breeding work due to the rapid rejection of plants with undesirable traits.

## 4. Materials and Methods

### 4.1. Plant Material

The following accessions of wheat were studied: 60 cultivars of winter durum wheat (Appendix A); 34 cultivars of spring durum wheat (Appendix A); 64 lines of winter durum wheat from the NGC’s CVT nursery (Appendix A); 41 lines of spring durum wheat from the NGC’s CVT nursery (Appendix A); 68 cultivars of winter bread wheat (Appendix A); 26 promising lines of winter bread wheat from the NGC’s CVT nursery (Appendix A). All lines from the NGC’s CVT nursery are stable, uniform lines of durum and bread wheats prepared for SVT and subsequent commercial use.

Additionally, 144 lines of winter durum wheat from a NGC’s breeding nursery with 1RS-translocated bread wheat in their pedigree were analyzed (Appendix A). These accessions were developed by crossing winter bread wheat carrying 1RS translocation with winter hard wheat with subsequent re-crossing resulted hybrids with durum wheat. Then, the hybrids were self-pollinated for two years and individual spikes were selected according to the following criteria: visual absence of diseases, high grain number per spike, absence of underdeveloped spikelets in the lower part of the spike, tillering, height, *leucurum* variety. Lines with the signs of plant damage after overwintering were selected in small quantities or rejected.

### 4.2. PCR Genotyping for 1RS Translocation

The genomic DNA was extracted from the dried leaves of adult plants as well as four-day-old seedlings as described in [68]. The presence of translocations involving 1RS in wheat background was detected using microsatellite marker SCM9: F: TGACAACCCCCTTTCCCTCGT; R: TCATCGACGCTAAGGAGGACCC [69,70]. Each 25-μL PCR reaction consisted of 70 mM Tris-HCl buffer (pH 9.3), 16.6 mM (NH_4_)_2_SO_4_, 2.5 mM MgCl_2_, 0.2 mM each dNTP, 30 pM forward and reverse primers (Sintol Ltd., Moscow, Russia), 0.05 U/µL Taq polymerase (Sileks, Moscow, Russia) and 4 ng/µL template DNA. Amplification was performed using Bio-Rad T100 (Bio-Rad, Hercules, CA, USA) with the following conditions: initial denaturation at 95 °C for 10 min followed by 45 cycles 95 °C for 30 s, 60 °C for 30 s, and 72 °C for 1 min and a final step at 72 °C for 10 min.

PCR products were separated in 2% agarose gel with TBE buffer (90 mM Tris-HCl, pH 8.3, 90 mM boric acid, 0.1 mM EDTA), using the GeneRuler 100 bp Plus molecular weight marker (Thermo Fisher Scientific, Madison, WI, USA) in Bio-Rad Sub-Cell horizontal electrophoresis chambers in conjunction with a PowerPac Basic power supply (Bio-Rad, Hercules, CA, USA). The gels were stained with ethidium bromide and documented using the Gel Doc XR+ system (Bio-Rad, Hercules, CA, USA) under ultraviolet light. The resulted 207 bp or 228 bp amplicon corresponds to 1RS.1BL or 1RS.1AL translocation, respectively.

An additional marker for 1RS.1BL translocation verification was the competitive allele-specific KASP marker 1B1R_6110 for SNP wMAS000011: F: GGAGCAGGTCCAGATCGCG; R: CGGAGCAGGTCCAGATCGCA; Common: GAAGCTCCGGTAGATGGAGGCTA (LGC, Teddington, UK) [71]. Amplification was carried out according to the manufacturer’s recommendations.

### 4.3. GISH Protocol

Chromosome spread preparations were made from root tip of three-day seedlings using the squashing technique, as described in [72]. Rye genomic DNA (50 ng per preparation) labeled with (DIG)-11-dUTP using Dig Nick translation mix (Roche, Mannheim, Germany) was used as a probe. The DNA of wheat cv. Moskovskaya 39 fragmented to 200–500 bp by autoclaving was used a block. The probe-to-block ratio was 1:20. Genomic in situ hybridization was performed as described in [73]. After hybridization, the chromosomes were counterstained with 1 mg/mL DAPI. Signals on the chromosome preparations were visualized and recorded using an AxioZeiss Imager M1 fluorescence microscope equipped with AxioCam MRm CCD camera (Carl-Zeiss, Oberkochen, Germany).

## Figures and Tables

**Figure 1 plants-12-01205-f001:**
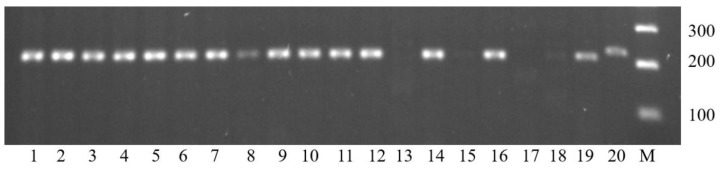
Results of 1RS detection in winter bread wheat cultivars using SCM9. 1, Kavkaz; 2, Kurs; 3, Kollega; 4, Vassa; 5, Zhiva; 6, Vita; 7, Ayvina; 8, Yuka; 9, Urup; 10, Antonina; 11, Vid; 12, Videya; 13, Liga 1; 14, Irishka; 15, Moskvich; 16, Ol’khon; 17, Zimtra; 18, Tvorets; 19, Bagrat; 20, 2318h9-4-28; M, GeneRuler 100 bp Plus DNA Ladder. Lanes from 1 to 12, 14, 16 and 19 show 207 bp amplicon corresponding to translocation 1RS.1BL; lanes 13, 15, 17 and 18 show no amplification corresponding to the absence translocation; lane 20 shows 228 bp amplicon corresponding to translocation 1RS.1AL.

**Figure 2 plants-12-01205-f002:**
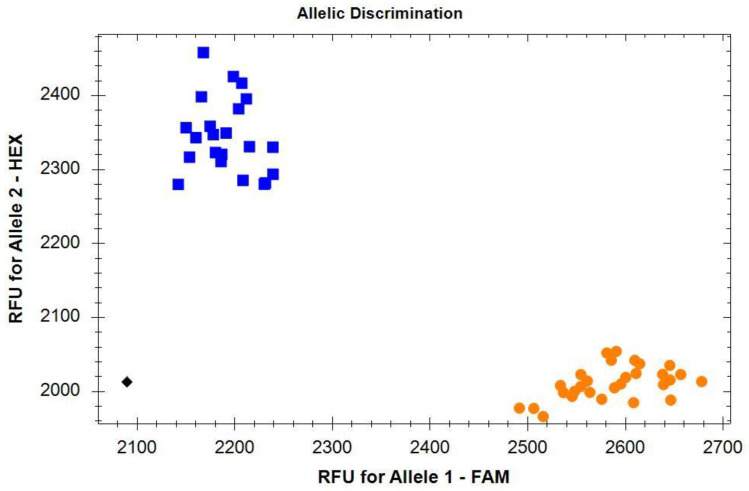
Application of KASP marker 1B1R_6110 for the detection of translocation 1RS.1BL. Blue squares (HEX axis), the presence of the 1RS.1BL translocation; orange circles (FAM axis), the absence of the 1RS.1BL translocation; the black rhombus in the lower left corner is the negative control (water).

**Figure 3 plants-12-01205-f003:**
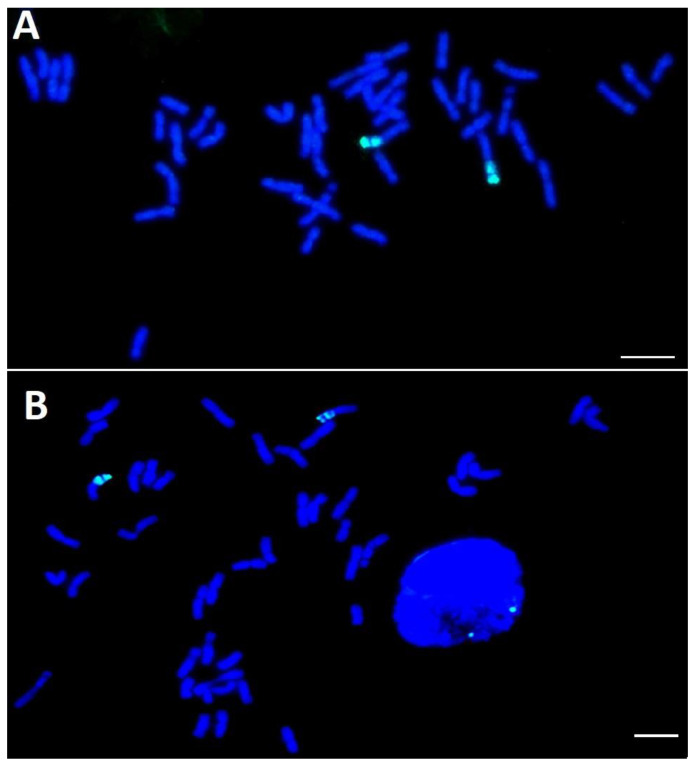
Genomic in situ hybridization on the chromosome spreads of winter bread wheat cultivar Vassa (**A**), and line 2318h9-4-28 (**B**). Green signals indicate rye chromatin, scale bar indicates 10 μm.

**Figure 4 plants-12-01205-f004:**
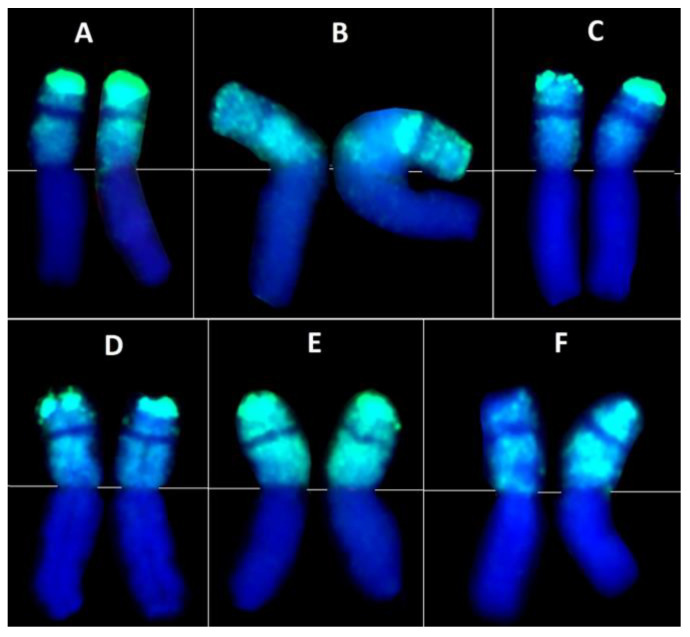
Centric rye-wheat translocated chromosomes revealed using genomic in situ hybridization in winter bread wheat cultivars Tanya (**A**), Avrora (**B**), Vassa (**C**), Polovchanka (**D**), Knyazhna (**E**), and line 2318h9-4-28 (**F**). Green signals indicate rye chromatin.

## Data Availability

Data are contained within the article and Appendix A.

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
