# Peer review of "Permanent Spreading of 1RS.1AL and 1RS.1BL Translocations in Modern Wheat Breeding"

_plants, 2023, doi:10.3390/plants12061205_

Round 1

Reviewer 1 Report

The manuscript reported a comprehensive PCR and GISH analysis of the cultivars and collections of bread and durum wheat from Lukyanenko National Grain Centre (NGC) , and found that the absence of 1RS translocations in the studied 22 durum wheat germplasm.  The results provide insight for the understanding the genetic diversity in NGC deposited wheat and durum wheat. The relevant genotyping results are helpful for the future wheat and durum wheat breeding practices. The study was clearly presented, and the conclusions were supported by multiple evidence. The GISH figures are in great high quality.

In general, the manuscript is good written, and it can be suitable for publication after considering the comments:

1.      The word in situ should be italic in the overall text.

2.      Lane 156-159, the description of the results of Fig. 3 and Fig. 4 should be expanded. The GISH of representative metaphase cells can be used to show the differences of 1RS.1BL and 1RS.1AL, and further confirmed the PCR results for the lines. Since the GISH can distinguish the homozygous of heterozygous of the translocation while the PCR does not.

3.      The scale of bars in Fig. 3 should be indicated in caption of the figure.

4.      Overall the References list, the scientific name of species must be italic.

Author Response

Dear Reviewer 1,

We appreciate the positive feedback from you and the helpful suggestions that you have made. We would like to thank you for the very constructive comments, which we believe will substantially improve the quality of the manuscript. We have been able to incorporate changes and reflect all of the suggestions you have provided. Please find below our point-by-point response to your comments and concerns.

Comment #1. The word in situ should be italic in the overall text.

Answer #1. According to your suggestion, the format of the word "in situ" changed to italics throughout the text.

Comment #2. Lane 156-159, the description of the results of Fig. 3 and Fig. 4 should be expanded. The GISH of representative metaphase cells can be used to show the differences of 1RS.1BL and 1RS.1AL, and further confirmed the PCR results for the lines. Since the GISH can distinguish the homozygous of heterozygous of the translocation while the PCR does not.

Answer #2.  According to your recommendation, we added the description and discussion of the results of Fig. 3 and Fig. 4 to "Discussion", paragraph 2.

Comment #3. The scale of bars in Fig. 3 should be indicated in caption of the figure.

Answer #3. According to your suggestion, we added the scale of the bar to the caption of Fig. 3. Additionally, we left only two metaphase plates, representing 1RS.1BL and 1RS.1AL, according to the suggestion of Reviewer 2.

Comment #4. Overall the References list, the scientific name of species must be italic.

Answer #4. According to your recommendation, we revised the reference list and changed the font of Latin names of species to italics.

Kind regards,

Authors

Reviewer 2 Report

Comments fort he eidtor & authors

# Title: should be modified:

“Permanent spreading of  1RS.1AL and 1RS.1BL translocations in modern wheat breeding

#  Chpt. Introduction

It should be reduced and focused on the issue of wheat breeding (see comments attached to the manuscript)

# Chpt. “Material & methods”

In the supplemented tables the wheats should be described more detailed, i.e., origin, breeder, release, growth habit, SNP markers etc.)

# Chpt. Results

# A table should be included, just showing the critical introgression wheats

 ## Fig. 1 shows wheat that are not mentioned on page 3, lines 123-127, why?

## A Table with all wheats showing the different types of introgressions would be helpful

# Chpt. References

Should be supplemented as given in the manuscript.

Author Response

Dear Reviewer 2,

Thank you for your time, effort, and positive feedback. Please accept our appreciation of your respectable view and valuable comments, which added a lot to the manuscript. All comments have been taken into consideration, and the manuscript has been renamed and revised according to them. We have highlighted the changes within the manuscript. Please find below our point-by-point response to your comments and concerns.

Comment #1 Title: should be modified: "Permanent spreading of 1RS.1AL and 1RS.1BL translocations in modern wheat breeding"

Answer #1. According to your suggestion, we changed the title of the study.

Comment #2 Chpt. Introduction. It should be reduced and focused on the issue of wheat breeding (see comments attached to the manuscript)

Answer #2. According to your recommendation, we moved two paragraphs about NGC and Krasnodar climatic conditions to "Discussion". Other minor corrections highlighted in the attached pdf file have also been accepted.

Comment #3 Chpt. "Material & methods". In the supplemented tables the wheats should be described more detailed, i.e., origin, breeder, release, growth habit, SNP markers etc.)

Answer #3. According to your suggestion, we added information about breeders and year of registration for spring durum, winter durum, and winter bread wheat cultivars in Supplementary Tables S1, S2, and S5; growth habit is indicated in the Supplementary Table captions. As for the other Supplementary Tables, they contain information about present-day breeding lines from competitive variety trials and breeding nurseries at the National Grain Centre, as we have additionally noted in the "Plant Material" and in the titles of Supplementary Tables for clarity.

Comment #4 Chpt. Results. A table should be included, just showing the critical introgression wheats

Comment #6 A Table with all wheats showing the different types of introgressions would be helpful

Answer #4 & #6. We are afraid of the manuscript being overloaded by an additional table that duplicates the information in the text. The 33 cultivars of bread wheat with 1RS.1BL are listed in the beginning of "Results", while the names of CVT lines with 1RS would be difficult to read. However, we understand your recommendation, and for clarity, in paragraphs 1 and 2 in "Results", we added references to Supplementary Tables, where a reader may find information on which accessions carry 1RS.

Comment #5 Fig. 1 shows wheat that are not mentioned on page 3, lines 123-127, why?

Answer #5. There were some technical flaws in the Figure 1 caption that we carefully revised and corrected. We are grateful for your attentiveness to such details.

Comment #7 Chpt. References. Should be supplemented as given in the manuscript.

Answer #7. According to your suggestion, we have incorporated all the references that you provided.

Comment #8 The best figure should be selected for demonstration e.g., Vassa (C). The remaining pictures can be removed!

Answer #8. According to your recommendation, in Figure 3, we left two metaphase plates , representing 1RS.1BL in cv. Vassa and 1RS.1AL in breeding line 2318h9-4-28., 

Comment #9 introduce a Chpt. "Material and methods" before Chpt. "Results

Answer #9. We would be glad to change the order of the chapters according to your reasonable recommendation. However, the manuscript is formatted according to the MDPI Plants template. If the editor has no objections, we will introduce "Materials and methods" before "Results".

Kind regards,

Authors